# “I Felt I Was Actually Inside the Movie” An Exploratory Study into Children’s Views of Staging a Physically Active Experience, with Implications for Future Interventions

**DOI:** 10.3390/ijerph20043624

**Published:** 2023-02-17

**Authors:** Emily Budzynski-Seymour, Michelle Jones, James Steele

**Affiliations:** 1School of Sport, Health, and Social Sciences, Solent University, Southampton SO14 0YN, UK; 2Sport, Physical Education and Coaching, Plymouth Marjon University, Plymouth PL6 8BH, UK

**Keywords:** physically active experience, themed entertainment, children, physical activity, affective response

## Abstract

Current levels of inactivity suggest novel approaches are needed to engage children in physical activity (PA), and enjoyment is a strong motivator for children’s PA engagement. A *physically active experience* (PAE) was proposed as a way to use entertainment, education, (e)aesthetics and escapist methods to promote PA to children in a way that is immersive and enables them to actively partake whilst enjoying their experience. In this current mixed methods study, three physically active experiences based on popular children’s movies were designed and staged, in order to explore children’s views on staging a PAE and provide implications for future PA interventions. Seventeen children (boys n = nine, girls n = eight) between the ages of nine and ten years provided feedback on the experiences. The children watched a pre-recorded video presenting the physically active experiences and then completed a survey including affective forecasting responses, which was followed by participation in an online focus group where views on the experiences were explored further. For all three experiences, the mean anticipated affective response for valence was between ”fairly good” and “good”, and for arousal between “a bit awake” and “awake”. Further, when asked, the children reported wanting to take part in the experiences (experience 1: 82.4%, experience 2: 76.5%, experience 3: 64.7%). The qualitative data revealed that children felt that they would enjoy the sessions, feel immersed in their environment, transported away from reality, and that they would be able to learn something new regarding PA. These results support the implementation of a PAE to engage children in enjoyable PA; future interventions should use these findings to engage children in a PAE, examining their actual responses to the activities.

## 1. Introduction

The World Health Organisation (WHO) recommends that children engage in at least 60 min of moderate to vigorous physical activity (MVPA) daily spread over the week [1]. Alarmingly, however, data suggest that children are moving less and are adopting more sedentary lifestyles, with the majority not meeting the physical activity guidelines [2,3]. As such, the negative health consequences of failing to be physically active, which are commonly reported in an adult population, are now increasingly being reported in children [4,5]. Enjoyment is a strong motivator for children’s physical activity engagement [6], a lack of which is often the reason for children to cease engagement [7]. Ensuring a positive affective response, which is associated with increased engagement in physical activity, can help to build and sustain motivation for physical activity over time [8]. If a greater understanding and emphasis on children’s levels of enjoyment and their affective response to physical activity is given, this may aid in the development of more effective physical activity interventions [7]. Indeed, recent studies have called for health promotion interventions that encourage enjoyable physical activities [9], and there has been research around alternative physical activity definitions that are more inclusive, holistic and which encompass the complex nature of physical activity, including the importance of emotive elements [10]. 

One recent strategy proposed to aid in promoting an engaging and enjoyable physical activity session for children is through staging a *physically active experience* [11]. This concept was based on the research around experiences [12] where there is an emphasis on engagement in the activity. When designing an experience, the environment is manipulated to “stage” the activity as an experience; the aim is to immerse those present in their environment with an emphasis on the overall experience offered [13]. There are four elements, known as the four Es, that need to be considered to create such an environment. These are: entertainment, education, (e)aesthetics and escapism. When staging a physically active experience, the aim is to make children feel swept up in the activity and immersed in their environment, at the same time as engaging in physical activity so that they enjoy their experience. Csikszentmihalyi and Robinson [14] explain that those who partake in an educational experience may *want to learn*; in an escapist experience *want to go and do*; in an entertainment experience *want to enjoy*; and those partaking in an (e)aesthetic experience just *want to be*. A physically active experience therefore is proposed to help children want to learn, want to go and do, want to enjoy and just want to be physically active. 

As detailed in the original paper [11], to ensure a *want to be* in the space, the environment where the activity is staged needs to be inviting. Examples suggested for how this can be done included the incorporation of a brand to help draw children in, by theming the environment so it is staged to look a certain way (e.g., an alien planet or deserted island), or by including music to help with the audio (e)aesthetics. The next step involves immersing children in the space and making them active participants by creating a *want to do*. Through incorporating characters and a narrative into the experience, children can adopt a role and follow a story, and both are key engagement tools [15]. The role adopted by children in the story can facilitate them engaging in specific fundamental movement skills, which links to the next stage—education, ensuring a *want to learn*. In the story, the character may need to jump on stepping stones to cross a river, so the children will engage in jumping. Finally, there is an overall emphasis on enjoyment—a need to ensure a *want to enjoy*. The combination of these elements is designed to ensure an enjoyable experience. In addition, children are allowed to use their imagination, which facilitates an increase in intrinsic motivation [16]. Additionally, all the experiences are game based, which again promotes enjoyment, as the sole purpose of games is to amuse or entertain [17]. 

It was suggested that future research is needed to explore the ideas presented around a physically active experience, in order to investigate potential benefits for children’s physical activity engagement. As such, this study aims to explore children’s responses to physically active experiences and, from this, provide guidance for the development of future physical activity interventions. 

## 2. Materials and Methods

### 2.1. Participants 

Seventeen participants (9 boys, 8 girls) between the ages of nine and ten took part in this mixed methods study. The children were from Year 5 (England), and pseudonym information can be found in Section A.1. One school in Hampshire, England, was used for this data collection (2019 school data: 352 pupils on roll, 13.6% eligible for free school meals, 8.5% English not first language). Written parental consent to participate was obtained and the children gave verbal assent. Ethical approval was gained from Solent University, Southampton. 

### 2.2. Protocol

It is important to note that this study was conducted during the COVID-19 pandemic. During the period of data collection, many social distancing rules were in place. As such, it was not possible for researchers to engage children in the activities in person, as in-person data collection was not allowed. Children therefore provided feedback around the idea of a physically active experience after watching videos provided by the researchers of three experiences in a mixed methods repeated measures study. There were three parts to each video: First, a short clip from the relevant film, which was used to set the scene and introduce the characters and narrative. The children were then given a tour of the themed space and shown the overall environment and any themed equipment used, with the music playing in the background. Examples of some of the themed equipment can be seen in Figure 1a–c. Finally, they were shown a short clip of one person taking part in the game’s activities to illustrate what the physical activity skill would involve. Three Disney films were chosen to base the experiences on (the incorporation of Disney was made under “fair use”, as this relates to its properties being able to be used for educational purposes). The Disney brand was chosen as a base for the experiences due to multiple previous interventions using their branding, including large-scale social marketing campaigns both in the US (e.g., VERB campaign) and the UK (e.g., Change4Life public health campaign). In addition, the Disney brand is known to be both very well-known and well-liked by children and suggested to have a positive influence over both affective responses and behaviours [18,19,20,21,22]. 

The videos (links provided in Section A.2) were embedded into an online survey for the first stage of data collection. This allowed for some demographical information to be collected from the children. After this, the children were shown the first video and asked a few questions. For each video, to help with fidelity, the children were asked if they had watched the whole video; they were also asked if they had watched the relevant films from which the experience was based. After that, the children were asked to imagine that they were about to complete the experience and asked how they think they would feel on a scale from very bad (−5) to very good (+5), as well as from very sleepy (1) to very awake (6). These were taken from the adapted versions of the children’s feeling scale (CFS) and children’s felt arousal scale (CFAS) by Hulley [23] and would provide an insight into the children’s anticipated affective response to the experiences—both valence and arousal. Unlike in adults, children tend to be accurate when it comes to affective forecasting [24]. Finally, the children were asked if they would like to take part in the experience. This was then repeated for the second and third experiences. 

As soon as the children had completed the survey, they then signed into a prescheduled online focus group. Focus groups were completed to gain a deeper understanding of the children’s opinions of the experiences. The questions were structured around the four Es of experience design—asking them about the educational, (e)aesthetic, escapist and entertaining aspects of the experiences. There were four focus groups in total; all had between four and five participants and were around 30 minutes long. To help promote a child-friendly atmosphere on the videocall, the background of the camera view of the researcher contained some of the themed equipment from the videos, and these were referred to and brought forward at different points. 

### 2.3. Materials 

All of the experiences were designed in line with the research presented on staging a physically active experience. Table 1 illustrates how each of these four components were addressed. 

#### 2.3.1. Video 1—The Jellyfish Jump Physically Active Experience 

The first experience that was staged was based on the Disney Pixar film *Finding Nemo* and was entitled “Jellyfish Jump”. The aim of the activity, or the educational element of the experience, was for children to practice their jumping skills. The narrative behind this game was taken from a particularly adventurous scene from the film, where Marlin and Dory (two of the main protagonists) find themselves in a swarm of jellyfish and need to escape. Marlin explains to Dory how to do this—they need to jump on the tops of the jellyfish as they will not sting them. Herein lies the game: the children take on the role of either Dory or Marlin and try to escape the jellyfish swarm by jumping on the tops of the jellyfish. The space where the experience was staged was themed to look like it was set under the sea; there was blue tinsel hanging on the walls, with jellyfish hanging from the ceiling. Music was also used to add to the audio (e)aesthetics and the immersive elements of the space; music was chosen from the film’s soundtrack. Finally, there was themed equipment: the tops of the Jellyfish that the children needed to jump on were represented by blue hoops with jellyfish tentacles added. 

#### 2.3.2. Video 2—The Lightning’s Laps Physically Active Experience

The second experience that was staged was based on the Disney Pixar *Cars* franchise and was entitled “Lightning’s Laps”. The aim of the activity, or the educational element of the experience, was for children to practice their running skills. The narrative behind this game was that the experience was staged as a racetrack, with race cars such as Lightning McQueen (the films main protagonist) needing to perform some practice laps for their next race. The children would either take on the role of Lightning, or one of the other cars from the film. A large poster was used to illustrate some of the popular characters that the children could take on the role of, with chequered flags, cones and a racetrack all used to theme the environment. There were also themed cones inspired from the films that the children could use to mark out their racetrack. Once the track was marked out, the children’s role was to run or drive their cars around as fast as they could. Music from the films was also used here: two upbeat tracks were played in the background. 

#### 2.3.3. Video 3—The Andy’s Coming Physically Active Experience 

The final experience that was staged was based on the Disney Pixar *Toy Story* franchise and was entitled “Andy’s Coming”. The aim of the activity, or the educational element of the experience, was for children to practice their balancing skills. The narrative behind this game was that one of the main characters in the film, Buzz Lightyear, needed help to collect and return three of the Alien toys from one side of Andy’s bedroom to the other. The children would take on the role of one of the characters from the film and help to bring the Aliens back to Buzz. However, just like in the film, whenever the toys’ owner appears the toys must stay still so that he does not realise that they are alive. There were footprints across the room for the children to follow and balance on if Andy were to appear—this was their opportunity to practice their balancing skills. The presence of Andy was represented by the background music stopping and someone shouting, “Andy’s coming”. In the environment, there were toys from the film and three oversized building blocks to represent the fact that the children were taking on the role of a toy, and thus, had been shrunk. There was also a large image depicting many of the toys in Andy’s room. 

### 2.4. Data Analysis

The focus groups were audio recorded and transcripts were written up verbatim with the children’s names substituted for pseudonyms for confidentiality. A thematic analysis was chosen, as it allows for the identification of patterns of meaning across a data set [25]. A deductive approach was used for this analysis, as the research from designing a physically active experience was used to analyse the data. The data collected from the children were categorised into four themes based on the 4 Es: education, (e)aesthetics, escapist and entertainment. The analysis process began with immersion in the data to gain familiarity. The higher level themes were established beforehand due to the deductive approach; however, codes were developed for each of these themes. Reviewing was then completed to ensure each theme and codes were coherent. Due to the reflective nature of this process, one researcher was used for the analysis; however, a second researcher provided some assistance, again in a reflective manner, e.g., sense checking. 

## 3. Results and Discussion

The results are presented and discussed in an exploratory way in relation to the four Es of staging a physically active experience, and they offer some evidence and support for its use in engaging children in physical activity. Suggestions are also provided based on these results and previous literature for how these findings may support the future development of physical activity interventions, noting the implications for future interventions that actually engage children in these experiences. 

### 3.1. Theme 1—(E)aesthetics 

The first theme was the *(E)aesthetics*, ensuring there was a *want to be* in the environment. Some example comments for this theme can be seen in Figure 2. An aim was to make children feel like they were in a scene from the movie and immersed in their environment. The data reflected this, as the children commented how they felt the environments “*looked a lot like the movie, and I felt I was actually inside the movie*” (Ava, girl). This was captured by the children expressing how they “*could imagine being in the film*” (Daniel, boy). It has been suggested previously that future children’s physical activity interventions should consider providing spaces and facilities such as castles, moats and foam swords to help the children feel swept up in their environment, potentially making it more inviting [26], which occurred within this study. Regarding the inclusion of environmental (e)aesthetics, “*if there was none then that would just be a bit boring*” (Emma, girl), illustrating how these additions help to ensure a want to be in the environment. Specific theming was also noticed by the children, for example: “*The fact that there’s a theme and it’s not just a bunch of hoops on the floor, there’s actually a theme to it*” (Noah, boy). Based on this and previous research, future interventions should carefully consider the space where the experience will be staged, ensuring it looks visually appealing to help draw the children in. 

An additional method of adding to the *(e)aesthetics* of the environment was the inclusion of music. It has been reported that music can help with motivation, control arousal, reduce feelings of exertion and improve mood, all contributing to an increase in exercise adherence [27]. Research around the benefits of audio–visual elements during a physical activity session with children has found that these elements can help with the reported affective response and sense of immersion in the activity [28]. Additionally, Vazou and colleagues [29] reported that a “novel” physical education (PE) class which included using upbeat engaging music and videos led to higher enjoyment compared to a “traditional” PE class [29]. In the current study, the children were asked their opinions around the inclusion of music and felt that: “*If you have music it would be better as it would take your mind off what you were doing and maybe keeps you going a bit longer*” (Noah, boy). This supports the research around music in terms of its ability to distract from feelings of exertion, allowing longer engagement in physical activity. Therefore, interventions aiming to implement a physically active experience in the future should note the importance of the audio (e)aesthetics and potentially incorporate this into their environments, as the findings from the current study agree with those from previous studies highlighting its benefit. 

### 3.2. Theme 2—Escapist

The second theme was the escapist element or ensuring a *want to do,* a summary is provided with example comments in Figure 3. Narratives are key engagement tools, as when children hear a story, they want to follow its events [30]. This was reflected in the data collected here. Each experience incorporated a narrative as a part of this escapist stage. The use of a story in the activity made it seem “…*a bit more lively and made me actually want to try it*” (Clara, girl). The children also seemed to find the fact that a Disney story was used particularly effective, “*Because it is one of your favourite films and you would think that you were actually in the film so you would be more active in it*” (Thomas, boy). This is supported by previous Disney-based research where the inclusion of this brand was noted to lead to specific benefits [31]. Further, Public Health England found that 64% of children reported that they would be more physically active if they saw their favourite Disney characters being active [32]. 

As well as a narrative, characters were also incorporated into the experiences, and the positives associated with this are present in the results. When characters are used in a story where children can identify with them, this can lead to increases in engagement and acceptance of the message being delivered [15]; in this case, increasing the *want to do* and be active. When asked about the inclusion of the characters, the children often reacted positively: “*It makes you feel like they are actually with you and that you are actually the characters, so the movie is basically with you*” (Ava, girl) and “*You are comparing it to the movie, so you are actually doing what the characters are doing*” (Clara, girl). Although characters and narratives have been incorporated in previous health research, their use in a child’s physical activity setting is limited. A study exploring Public Health England’s Change4Life 10 min Disney-branded Shake Ups [31] included qualitative views from parents. In relation to characters, a parent commented that their child “*enjoyed having specific roles to play*”. The current research aligns to the previous research where the children felt having a narrative or story would mean they believed they would “*be more active in it*” (Thomas, boy). Future interventions could consider the use of a narrative and characters that is engaging and will resonate well with the children, ensuring a want to escape and be transported into the story, taking part in the character’s journey. 

### 3.3. Theme 3—Educate

This third theme relates to engagement in the specific physical activity skills and ensuring a *want to learn,* see Figure 4 below. As Pine and Gilmore [33] state, although education is serious business, that does not mean that educational experiences cannot be fun. Incorporating fun into an experience might enable it to become a much-desired learning opportunity, and research in a physical activity setting has shown that memories of previous physical activity experiences such as PE lessons can influence future engagement [34]. The children were asked about the physical activity aspect of the experiences, with comments including “*I liked how we had to jump into the different hoops*” (Chris, boy) and that they liked a specific game because “*it has a lot of physical movement and stuff*” (James, boy). Comments also supported the ability of these experiences to ensure a *want to learn* by providing an educational experience, but in a fun way: for example, it would “*be good exercise*” (Sarah, girl), and that they felt they would be able to “*get better and progress but you could do it in a fun way*” (Liam, boy). The skills were woven into the narrative with the characters; because of this, the children were able to practice them in a more fun and engaging way where they could use their imagination. This is supported by comments from the children, including that they felt “*Happy I had done something physical for a while in an imaginary world*” (Noah, boy), and the reason that they liked the Jellyfish Jump game was “*because it was fun to jump in the hoops and to avoid the jellyfish*” (Ava, girl). This also illustrates how the skill was woven into the narrative. Finally, the physical activity actions linked to the narrative and characters were also considered “*cool and that kinda made me feel like I was in the movie which honestly makes a difference*” (Clara, girl). It is suggested that future interventions could benefit from carefully considering how to weave the narrative and story with the skills they want to promote, or the activity they would like children to engage in. If this is carried out in a creative way, it might be expected that children will be able to learn in a more enjoyable way. 

### 3.4. Theme 4—Entertainment

The last theme was the entertainment theme, ensuring an enjoyable experience and a *want to enjoy*. This relates to ensuring a positive affective response during the experiences. During the focus groups, the children were asked to imagine that they were about to enter the experience and think about how they would feel. Research suggests that children are more accurate than adults at affective forecasting [24]. The most common feeling supporting this theme was excitement: “*I’d feel quite excited to play…like I would enjoy myself*” (Mia, girl). The children were also asked how they think they would feel when they were actually playing the game, expressing that they would feel “*A bit like, content that I was playing, like I was enjoying myself*” (Sarah, girl). This supports previous literature where studies have shown that children find activities where they are physically active fun [35]. Children also commented that they felt they would feel awake or prepared to play, for example, “*Ready and pumped up*” (Thomas, boy), “*Feel quite awake*” (Ava, girl) and “*I’d feel energised*” (Lucy, girl). The themes from the qualitative analysis support the quantitative data. For all three games, the mean (± standard deviation) anticipated affective response from the sample for valence was between “fairly good” and “good” (Jellyfish Jump: 1.8 ± 2.8, Andy’s Coming: 1.7 ± 2.7, Lightning’s Laps: 1.1 ± 2.7), and arousal was between “a bit awake” and “awake” (Jellyfish Jump: 4.4 ± 1.6, Andy’s Coming: 4.2 ± 1.3, Lightning’s Laps 4.7 ± 1.3). Further, the majority of children answered that after watching the video of the experiences, they would like to take part (Jellyfish Jump: 82.4%, Andy’s Coming: 76.5% and Lightning’s Laps 64.7%). Figure 5 presents some comments from the children in relation to this final stage of the experience. 

Linking to the physically active experience model [11], a key element for this final stage is the inclusion of games, with the children feeling that their inclusion was positive: “*Because you are in a game, you enjoy it a bit more*” (Thomas, boy). Indeed, evidence has shown that using games in a PE session can be an effective strategy and, due to the more active nature of game play, children are more likely to meet their MVPA recommendations when playing games [36]. Many games are considered more fun if they include elements of imagination [37], which is already noted as an important theme; therefore, if game play and imagination are used together, greater benefits may be achieved. This is evidence to support that by including these additional elements in the entertainment stage, children feel these would lead to them enjoying the experience more. This also demonstrates how a physically active experience could meet the need raised to encourage more enjoyable physical activities in children [9]. Future interventions should consider this final E carefully; the data shown here in line with previous research shows potential benefits from incorporating elements of imagination and games into a physical activity intervention. 

### 3.5. Negative Comments/Areas of Improvements

It is important to note that for each theme, there were comments that were either negative or suggested areas of improvements, some of these are presented in Figure 6. These were minimal: only 7.5% of the comments fell into this category. Although some analysis and reflection into these comments is presented here, it is also important to consider that these may be enduring negatives, drawbacks or limitations of a physically active experience; thus future research should consider these in any future experiences designed. One of the comments that was included was that “*I would feel kinda bored…because it is very repetitive just jumping*” (James, boy). This related to the Jellyfish Jump game. In the video, four hoops are visible and they are all positioned fairly close together; this could have been the reasoning for this comment. The researcher therefore proceeded to ask whether if more hoops were added, of all different sizes, and the route changed, would this be better? The child then commented “*then it would be more exciting and not repetitive*”. The full idea of the experience was hard to capture in the film due to the COVID-19 limitations. However, as seen in this example, if the space was expanded and more options available then the experience would be perceived as more exciting for the children. 

One child also noted that they felt that one of the games “*looks a bit childish*” (James, boy). Clearly, there are some characters that are focused at a younger audience and some that may be more suitable for an older audience. Interestingly, when asked what other films the children may like to be made into experiences, a common answer was the Marvel films. Marvel films have been used previously to promote physical activity in children, both by being incorporated into a social marketing strategy [38] and in a physical activity video using the Marvel characters [28]. It could be suggested that offering multiple experiences may be a good idea for future interventions, as this would allow the children to choose. Additionally, perhaps by offering a Marvel-based experience, which could be considered less childish, these comments on age suitability would be addressed. As stated by Pine and Gilmore [39], staging a series of experiences produces a more long-lasting effect on those attending, so multiple experiences and options may be a good idea. 

A final point was some negative comments around the use of music. Although music was discussed in the first theme, where the supportive comments of children were included, some children did not view the addition of music as positive. The views from this sample were mixed. One child stated that they would feel “*weird…. because I would hear music coming out from somewhere*” (Arlo, boy), and when they were asked if they felt music should or should not be included they stated “*shouldn’t be*”. In the *(e)aesthetics* section was one comment from a child who found the distracting nature of the music positive, which is in line with previous research [40]. However, another child did note that they “*didn’t really like the music as you were meant to get the toys and focus on that one thing, but you were kinda focusing on the music as well*” (Liam, boy), suggesting that they did not find the distraction helpful. Limited research is available on using music in a children’s physical activity setting, with some supporting its use [29], but due to the mixed nature of the comments included in this research, future research should investigate this further.

### 3.6. Limitations and Future Directions

Despite the important insights this research provided into the staging of a physically active experience for children, there are some limitations. Firstly, this study was unable to actually stage a physically active experience, allowing children to take part in person. Due to COVID-19 limitations, the children had to be shown a film and have the concept explained to them, followed by answering a series of questions based on what they had watched. The results presented therefore need to be interpreted in this context. They reflect what children anticipated they would feel, and what they thought of the video they were shown. These may reflect what they would actually feel and think; however, future less exploratory research needs to ascertain whether this is the case. Encouragingly, however, despite adults often exhibiting an overestimation bias in affective forecasting, children tend not to [24], suggesting that the activities would likely be enjoyed as reported. Again, due to COVID-19 limitations, the films that the children were shown were of a small, themed space. When discussing the experiences in the focus groups, the children were asked to imagine that a larger space—such as their school hall—had been themed in this way; however, this was not something that was able to be carried out at the time. For similar reasons, the filming only showed one participant playing the game, whereas in reality experiences are meant to be enjoyed by multiple children at once. Despite these limitations, the children’s overall responses to the experiences were positive, suggesting that if these were scaled up and children were given the opportunity to actually engage in the session, similar positive results would be expected. There is also a potential “novelty effect”, which was discussed in the original physically active experience paper [11]. As an example, research into using playground markings has shown a short-term increase in children’s physical activity levels; however, their long-term influence is less positive, and re-painting is suggested as a way to re-ignite enthusiasm [41]. This could mean that after a while, the characters and theming would need to be re-imagined, ensuring there were still high levels of engagement by ensuring a continuous “novelty effect”. 

## 4. Conclusions

This study provides a deeper insight into the proposed ideas around designing and staging a physically active experience, by collecting mixed methods feedback from children as to their thoughts on a Disney-themed experience. Overall, the data support this notion that a physically active experience may have a positive influence over children’s engagement in physical activity. When discussing the results in relation to the four Es, there is potential for these experiences to ensure a want to be, a want to do, a want to learn and a want to enjoy. However, these results must be interpreted giving consideration to the limitations of this study, including its exploratory nature and the COVID-19 based limitations. Future physical activity interventions should take this research and the suggestions it provides and design, implement and evaluate children’s actual engagement in these experiences. This will allow for an identification of their feelings and opinions of the experiences, rather than the predicted responses reported here. In conclusion, the findings from this study further the research around a physically active experience and demonstrate its potential in being an effective, new and novel way to engage children in physical activity.

## Figures and Tables

**Figure 1 ijerph-20-03624-f001:**
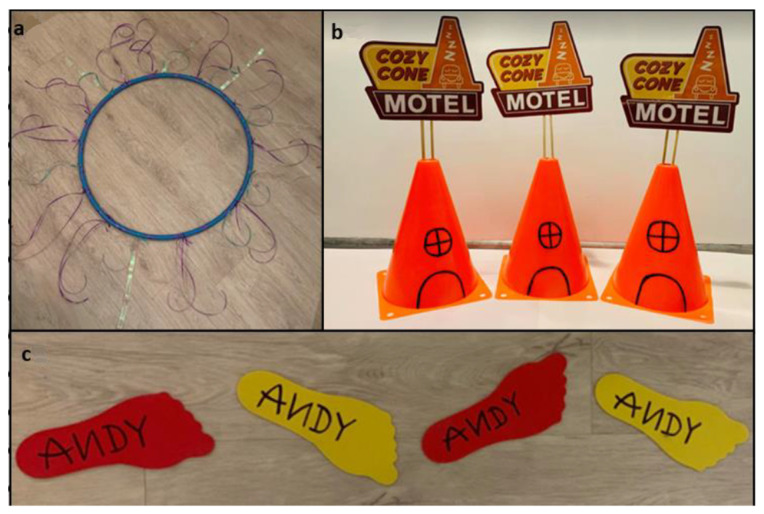
Themed equipment: (**a**) jellyfish hoop from Jellyfish Jump, (**b**) cones from Lightning’s Laps, (**c**) footprints from Andy’s Coming.

**Figure 2 ijerph-20-03624-f002:**
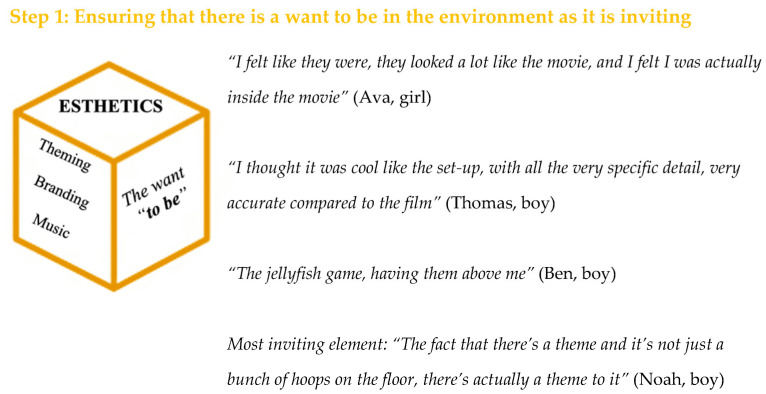
(E)aesthetics results.

**Figure 3 ijerph-20-03624-f003:**
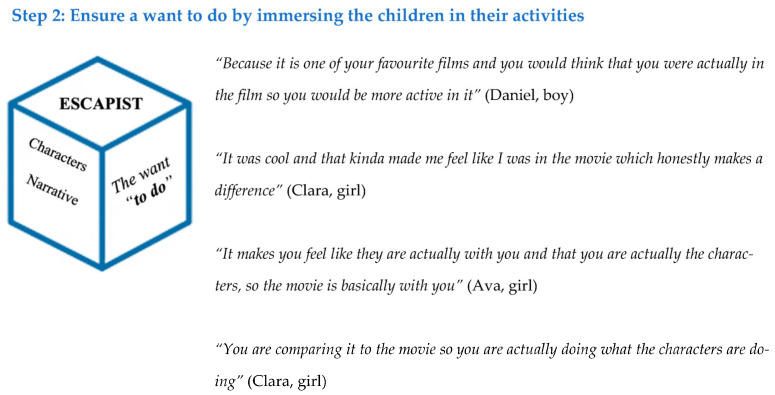
Escapist results.

**Figure 4 ijerph-20-03624-f004:**
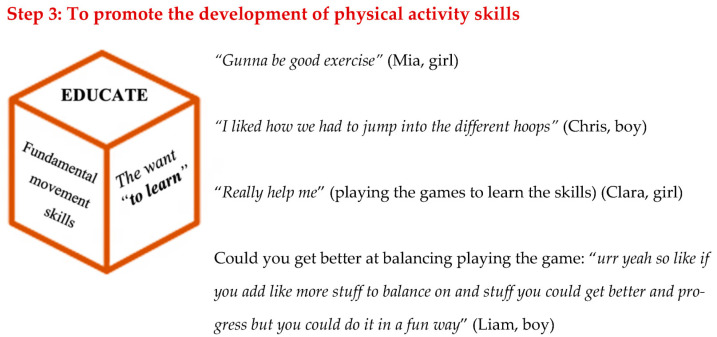
Educate results.

**Figure 5 ijerph-20-03624-f005:**
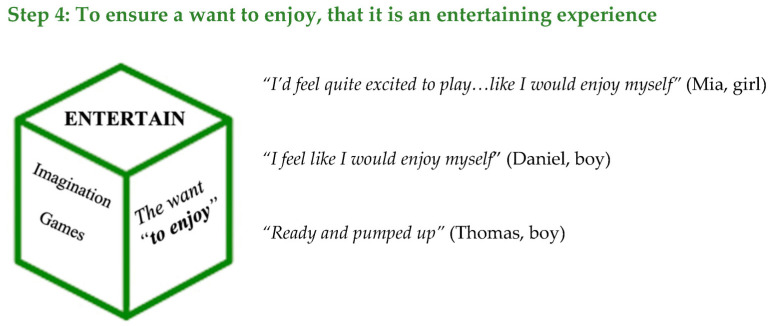
Entertain results.

**Figure 6 ijerph-20-03624-f006:**
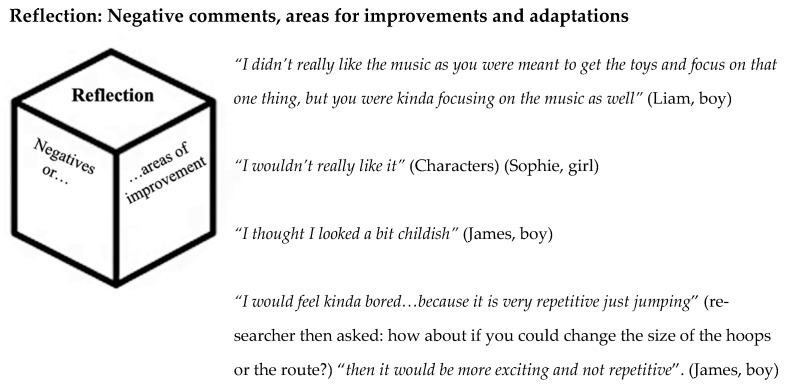
Reflection.

**Table 1 ijerph-20-03624-t001:** Addressing the 4 Es of a Physically Active Experience.

(E) AESTHETICS:
Each space was themed to look like a specific scene from a popular children’s movie, e.g., under the sea, a racetrack or a bedroom full of toys. The Disney brand, as justified, was incorporated to aid with the theming: the space was themed to a specific film, e.g., *Finding Nemo*, *Cars* and *Toy Story*. Music played in the background to aid with the audio (e)aesthetics.
ESCAPIST:
A story was a huge part of the activity. The children were transported into the narrative, e.g., they were a race car training for a big event. Roles were given so children could escape into the stories, e.g., they were either Marlin or Dory and needed to escape the swarm of jellyfish by jumping on the tops, which were represented by the hoops.
EDUCATE:
Skills were woven into the narrative and movements of the characters, e.g., jumping, running and balancing. These skills needed to be completed to move the story and experience forward.
ENTERTAIN:
The children were encouraged to use their imagination, pretending they were taking on the role of the character in the story. The activities were game focused, playing towards a final result where the motivation was to be entertained and have fun.

## Data Availability

Data sharing is not applicable to this article.

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
