# Peer review of "“I Felt I Was Actually Inside the Movie” An Exploratory Study into Children’s Views of Staging a Physically Active Experience, with Implications for Future Interventions"

_ijerph, 2023, doi:10.3390/ijerph20043624_

Round 1
Reviewer 1 Report
Overall
I want to commend the authors for their work on this manuscript. It provides insights into holistic and novel approaches to engaging children in physical activity through so-called physically active experiences (PAE). As the authors are designing and staging three PAEs, the manuscript provides a promising effort to articulate and bring attention to children’s perspectives and interests in the development of future PA interventions. Overall, the manuscript is well-written and resonates with the aim of the journal. However, some comments and questions are below:
Abstract
P. 1, lines 8-26: For consistency and clarity, consider rewriting the abstract to include specific information about the three experiences design. Either by relating them to the four E’s or by writing that they are based on movies. This could help the reader understand what it means to design and stage three PAEs and how the term is used in the manuscript. Would also make it easier to read the introduction.
Introduction
The introduction does a good job of positioning the need for such research. A concern could be, however, that apart from the authors' research, the included research is mostly ten years or older. A suggestion could, therefore, be to position the manuscript in contemporary research that advocates for the holistic perspectives of PA and its relation to the environment. For instance, by considering the work of researchers such as Joe Piggin and Thiago Sousa Mathias. (The unifying theory of physical activity, 2022 and what is Physical Activity? A Holistic Definition for Teachers, Researchers and Policymakers, 2020).
The introduction (P. 2, lines 80-84) is ended with the aim of the manuscript, which is to ‘explore children’s responses to three physically active experiences and provide guidance for the development of future physical activity interventions implementing a physically active experience’. The manuscript could benefit from clarifying what is meant by the three PAE in the aim. Otherwise, a concern is that the aim is becoming abstract and risks not giving specific meaning to the second part of the aim.
Methods
Main concern: line P. 4, 141 – 183: The authors of this paper are ‘designing and staging’ three experiences to ‘guide the development of future PA (…)’. As such, a concern is to encourage the authors to emphasise how the four E’s (educational, escapist, entertaining and (e)aesthetic) are connected to designing the three experiences. This would help the transparency of the design and assist the reader in understanding the intentionality of the PAE’s design. In parallel to the discussion section, a clearer ‘design and staging’ description of the three PAEs would help understand surprising and unexpected patterns in the data.
P. 4, lines 185-191: For the qualitative deductive analysis the authors are referring to Braun (2016). I would, however, recommend reading the most recent work of Clark and Braun (2019, 2020, 2022) where they detail the deductive-inductive continuum. A second concern about the analysis is the transparency of why the sample quotes were chosen. Figure 2. could benefit from a further explanation. It would, for instance, help to give names to the four themes in figure 2, as this would help to understand what is meant in the sub-themes and match with the headlines in the discussion. In line with the tenets of Clarke and Braun, I would also suggest changing the substituted names such as “child 1” and “child 2” to pseudonyms. In coherence with the title of the manuscript, this could engage the reader in the quotation and further, the discussion.
P. 4, lines 192-200. In the result section, the PAE is referred to as ‘games’. For consistency, make sure that the language aligns.
Discussion
P. 6/7. Lines 209-214: The first section in the discussion could be rewritten for clarity. For instance, why are the result and discussion sections merged? In addition, I am mindful of how the authors of this manuscript provide suggestions for future interventions (p 6/7 lines 211-214). Does this mean that suggestions are provided based on their own experiences, or is it based on empiricism and theory? Although lines 227-229, 245-248, and 273-277 might indicate the latter, a suggestion for clarity and coherence would be to emphasise that suggestions are based on the data and supported in previous literature. Still, phrases such as “PAE in the future should note” and “should consider” mark attention for the authors to consider the data’s generalizability. Is it so that future research “should” or could it be “based on the results in this manuscript, future research may take into consideration”? My suggestion would be to either use formulations that do not assume generalizability, something which can be found on page 8, lines 299-302 and to some degree on page 9, lines 330-332, or to carefully consider rewriting the result and discussion section by dividing the implications into their sections.
Based on the data the authors provide in this manuscript, it would have been interesting to elaborate on the tension and complexity between the children by writing the results as a “story” with more in-depth information, instead of providing quotations and thereafter discussing it.
Conclusion
On page 10, lines 280-383 it is stated that “this study was unable to actually stage a PAE”, whereas in the conclusion it is stated that “this study has provided a deeper insight into the staging of PAE”. Although they may not be contradictory, a suggestion would be to focus on the design of the PAE as opposed to the staging. Finally, I would urge the authors to emphasise in the conclusion that the manuscript's findings must take into consideration the limitations of the study.
Reviewer 2 Report
Thank you for inviting me to review this study. I think the physical activity among children and adolescent is an essential area of research. This study is a small pilot study regarding scenarios to engage children in physical activity. Unfortunately, the covid-19 prevented the authors from offering the children to try the scenarios themselves, and instead offering them a video of the scenario. I think that this reduces the novelty and the value of this study.
Major remarks:
The presentation of the participants could be clarified. The authors describe the school setting, but was any of the participants non-english speaking an so on. This is more important than specifying which area the school was placed, in order to not get in any ethical problems.
I think that the qualitative data analysis must be described in detailed. It is not possible to understand how it was done.
I do not think that the table in the result display any results. It gives no clue to what the result really are. The sub-themes are no themes only consists of parts of the children´s quotes. The authors need to analyse the qualitative data and write a thematic text under each sub-theme explaining the content, and illustrate their interpretation with the quotes. I also think that both positive and negative should be in the same sub-theme, illustrating the variation of experiences.
The whole discussion part contains new result in form of not analysed quotes from the children. It is easy to think that the finding follows the discussion and not as it should be the other way around.
In the background and the method several choices has been described, like choosing Disney themes and in the discussion this is highlighted as a result, not bringing the knowledge further forward. In addition, the discussion around the different themes of the four E´s, repeat the theory. To save space this could be shortened and after re-analyzing the result, the theory could be better connected to the result
The addition of game in the end of the discussion is not all evident in the result.
It is a weakness that the negative responses from the children is discussed by themselves and that the authors have tried to explain why they occur. I do not think that there is a solution that could fit all, and learning from the different views of the children is a strength, which the authors acknowledge but could be handle in a better way. And doing a new analysis of the result could perhaps solve this matter.
Small remarks:
Lakshman 2010 is not present in the reference list.
Overall assessment:
Even though I think the authors have a lot of work left to get this study ready for publication, I hope they are willing to provide the work. I think it is an interesting area and all efforts to get children more physical active should be acknowledged.
Reviewer 3 Report
Thanks for the opportunity to review this paper. This paper addresses a very important topic of the literature regarding the physical activity and recreational opportunities for children. In spite of its merit and relevance to the field, the manuscript is not appropriate for publication as is. The manuscript requires significant editing before it can be ready for publication. The observations or comments to follow are made as suggestions to strengthen the manuscript and enhance its readability.
Abstract
The abstract seems like the authors are talking about a different parent and not about the current study
Introduction
I believe the authors needs to make a clear distinction about these experiences and the experiences of children in a typical physical education class. The purpose of the study it’s not clear as written.
Method
The authors claimed that they are using a mix-methods design. What type of mix-method? How the data is talking among themselves (quantitative and qualitative)?
Participants: I think additional information about the participants is needed. You indicated their age, but how about their race? Income level? What are their pseudonyms? This information is important when interpreting the results section.
Procedure: the link including the videos it’s not working. I could not see them to have a better understanding of the experiences. How long it will take for a physical activity professional to set the stage to recreate such experience? What’s the cost of the equipment to organize the activity?
Results: I think the results section is incomplete. The authors would be better off if they combine the results and discussion section together. The authors presented quotes from the participants, but we don’t know which child is saying what? Was a boy or a girl? How old was the child making the comment? A table with participants’ descriptions would be helpful
Discussion: the title says: Implication for Future Interventions. What are those implications? What’s the take home message of this study?
Round 2
Reviewer 2 Report
I think that the authors have improved the result/discussion section, which should be presented as only one headline "Result and discussion". Now it is possible for the reader to understand more of the result.
Author Response
Thank you for the comments and agreeing with our new "results and discussion" section, we are glad this has improved the manuscript and will be easier for the readers to understand